# LEAN-STAR:
# LEARNING TO INTERLEAVE THINKING AND PROVING

**Haohan Lin**[2][*]                    **Zhiqing Sun**[1]

**Sean Welleck**[1]                    **Yiming Yang**[1]

[1]Language Technologies Institute, Carnegie Mellon University
[2]Institute for Interdisciplinary Information Sciences, Tsinghua University

`{haohanl,zhiqings,swelleck,yiming}@cs.cmu.edu`

`https://leanstar.github.io/`

## ABSTRACT

Traditional language model-based theorem proving assumes that by training on a sufficient amount of formal proof data, a model will learn to prove theorems. Our key observation is that a wealth of *informal* information that is not present in formal proofs can be useful for learning to prove theorems. For instance, humans think through steps of a proof, but this thought process is not visible in the resulting code. We present Lean-STaR, a framework for training language models to produce informal thoughts prior to each step of a proof, thereby boosting the model's theorem-proving capabilities. Lean-STaR uses retrospective ground-truth tactics to generate synthetic thoughts for training the language model. At inference time, the trained model directly generates the thoughts prior to the prediction of the tactics in each proof step. Building on the self-taught reasoner framework, we then apply expert iteration to further fine-tune the model on the correct proofs it samples and verifies using the Lean solver. Lean-STaR significantly outperforming base models ($43.4\% \rightarrow 46.3\%$, Pass@64). We also analyze the impact of the augmented thoughts on various aspects of the theorem proving process, providing insights into their effectiveness.

## 1 INTRODUCTION

Theorem proving is a fundamental aspect of mathematics, and mathematical reasoning is an important part of artificial intelligence (Newell & Simon, 1956; Zhou, 2023). *Formalized mathematics* in particular provides a challenging testbed for assessing mathematical reasoning capabilities. Since theorems and proofs in this setting can be represented in the form of checkable source code, it is easy to evaluate proofs of arbitrary complexity (De Moura et al., 2015). Automated theorem proving, if successful, can also help discover unknown errors in previous proofs[1], and make it easier to guarantee that new proofs are correct. More broadly, formal mathematics coupled with powerful automation may unlock new forms of education and collaboration, mathematical insights, and applications to verifying critical software (Avigad, 2023; First, 2023; Buzzard, 2024; of Sciences, 2023).

Recently, language models have shown promising progress in formal theorem proving (Polu & Sutskever, 2020; Rabe et al., 2020; Wu et al., 2021; Han et al., 2021; Lample et al., 2022; Yang et al., 2023; Li et al., 2024). Existing approaches typically train a model solely based on the proofs in a formal language (code) such as Lean (De Moura et al., 2015), Isabelle (Nipkow et al., 2002), or Coq (Coq, 1996). Our key observation is that such approaches ignore a wealth of *informal* information that may be useful for learning to prove theorems (Welleck et al., 2021; 2022). For

---

[*]Work done during the visit at CMU.
[1]For example, Terence Tao found a non-trivial error while using Lean to formalize a project (Tao, 2023).

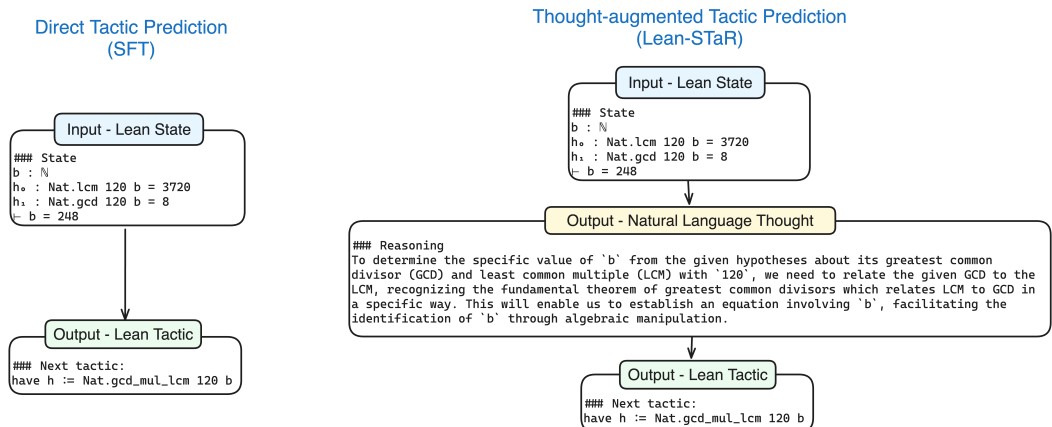

Figure 1: The illustration of tactic prediction in one proof step with and without thought.

instance, the underlying *thought process* prior to each step of a proof is not present in formal source code. Based on this insight, we propose to train a language model that can produce a natural language chain-of-thought ("thought") prior to each step ("tactic") of a formal proof.

We introduce Lean-STaR, a framework for learning to interleave informal thoughts with steps of formal proving. Building on the Self-Taught Reasoner (STaR) framework (Zelikman et al., 2022), we enable language models to interleave step-by-step rationales (i.e., thoughts) (Nye et al., 2021; Wei et al., 2022) with formal proving in a two-stage process. In an initial phase, we prompt a sufficiently capable language model, such as GPT-4 (Achiam et al., 2023), and generate retrospective thoughts based on a dataset of human-written proofs, such as Mathlib, the largest collection of human-written proofs in Lean (mathlib Community, 2020). Subsequently, we fine-tune a thought-augmented tactic predictor (Bohme & Nipkow, 2010; Blanchette et al., 2016; Gloeckle et al., 2023; Czajka & Kaliszyk, 2018) that, given a Lean state, can generate a thought and predict the subsequent tactic. In a second phase, we optimize this thought-augmented tactic predictor with the expert iteration algorithm (Anthony et al., 2017; Singh et al., 2023), using multi-step success rate in theorem proving as the reward.

Our work presents a new link between informal and formal mathematics, complementary to prior explorations that translate standalone mathematical statements (Szegedy, 2020; Wang et al., 2020; Wu et al., 2022) or translate informal proofs into formal proofs (Agrawal et al., 2022; Jiang et al., 2022; Azerbayev et al., 2023a; Zhou et al., 2024a; Huang et al., 2024). Lean-STaR generates natural language thoughts specifically for each proof step, improving formal proving capabilities by interleaving natural and formal languages.

We instantiate Lean-STaR by generating roughly 50,000 thought-augmented examples from Lean's Mathlib (mathlib Community, 2020), then synthesize an additional 50k examples through two iterations of expert iteration. To the best of our knowledge, this yields the first thought-augmented dataset for theorem proving. After fine-tuning an InternLM2-7b base model (Ying et al., 2024) on our thought-augmented data, our final Lean-STaR model can solve $34.8\%$ (pass@32) or $36.1\%$ (pass@64) of the problems on miniF2F-test (Zheng et al., 2021). Using stronger base model InternLM2-7b-plus, Lean-STaR can achieve $45.4\%$ (pass@32), significantly surpassing the previous results of $43.4\%$ (pass@32). In summary, Lean-STaR offers a framework for teaching language models to interleave informal thoughts with formal verification, advancing the capabilities of language models in automated theorem proving.

## 2 RELATED WORK

**Automatic Theorem Proving & Autoformalization.** Previous work on learning-based theorem proving typically follows the GPT-f framework (Polu & Sutskever, 2020), which trains a language model on (proof state, next-tactic) pairs, then proves theorems by using the model within a best-first tree search. Subsequent work has explored several directions, including data augmentation (Han

et al., 2022), novel proof search methods (Lample et al., 2022; Wang et al., 2023b), further training through curriculum learning (Polu et al., 2022), retrieval augmentation (Yang et al., 2023), or practical tools (Welleck & Saha, 2023). Others use prompted models to generate tactics (Azerbayev et al., 2023b; Thakur et al., 2023), or fine-tune models to generate a full proof (First et al., 2023). A second *auto-formalization* (Wu et al., 2022) thread incorporates informal mathematics into formal theorem proving. Draft-Sketch-Prove (Jiang et al., 2022) shows that language models have some ability to use informal proofs to improve a model's formal proving abilities, by drafting an informal proof, translating into a formal proof sketch, then completing the proof with tools like Sledgehammer (Bohme & Nipkow, 2010). Draft-Sketch-Prove and related methods (Wang et al., 2023a; Zhao et al., 2024; Zhou et al., 2024b) are limited to the Isabelle prover, since they use powerful automatic proving tools like Sledgehammer. Lean lacks these tools, so generating the entire proof at once would be more unlikely in Lean. We focus on Lean, and train language models to generate a thought and predict the subsequent tactic in each proof step. To the best of our knowledge, we are the first to introduce thought-augmented reasoning in automatic theorem proving.

**Rationale-augmented Reasoning.** Recently, many works demonstrated that letting language models reason before an answer can improve their performance on tasks including math, science, and code (Nye et al., 2021; Wei et al., 2022; Chen et al., 2022). Although the corresponding techniques (e.g., Scratchpad and Chain-of-Thought) have proven to be effective, they require either extensive annotated training examples or exposure to numerous similar examples during pre-training (Brown et al., 2020). The scarcity of natural language reasoning in formal theorem proving, coupled with the impracticality of manually annotating rationales for formal mathematics, thus presents a challenge. We propose a new Lean-STaR framework for *synthesizing* training examples by taking advantage of the correctness signal from the formal system.

**Bootstrapping Language Model Reasoning.** Recently, several works suggest that language models may be taught to reason via synthetic data that they generate themselves, akin to a reinforcement learning method that improves a policy through self-play. Polu et al. (2022) showed that a simple RL algorithm, expert iteration, paired with curriculum learning can improve a formal theorem proving model. Self-Taught Reasoner (STaR) (Zelikman et al., 2022) showed that we can iteratively fine-tune the language model on the correct (reasoning, answer) pairs generated by itself to gradually improve performance. Singh et al. (2023) proposed ReST-EM, which filters data generated by language model with a binary feedback signal rather than using fully manually annotated data (similar to expert iteration in (Polu et al., 2022)). Our work builds on these ideas, providing the first study of bootstrapped thought-augmented proving.

## 3 OUR METHOD: LEAN-STAR

We introduce Lean-STaR, a new method for combining informal thoughts with formal theorem proving. First, we recap interactive theorem proving (§3.1). Then we present Lean-STaR's data-generation (§3.2.1, §3.2.2) and reinforcement learning (§3.2.3) phases. Finally, we present our evaluation protocols (§3.3).

### 3.1 PRELIMINARIES

*Interactive Theorem Provers* (ITPs) are typically used for step-by-step automatic theorem proving in formal mathematics. At each step, we can provide the ITP with a high-level "tactic" to simplify the current goal state (e.g., the initial goal theorems to be proven) into subgoals. These subgoals will form new states, and proving all the subgoals results in a complete proof of the given theorem. We use Lean (De Moura et al., 2015), a popular interactive theorem prover. An example formal proof in Lean and its explanation are shown in Appendix D.

### 3.2 DATA GENERATION & TRAINING

We describe the data generation and training of the direct tactic prediction model (SFT), the thought-augmented tactic prediction model trained with synthetic data (Lean-CoT), and the final model trained with expert iteration (Lean-STaR).

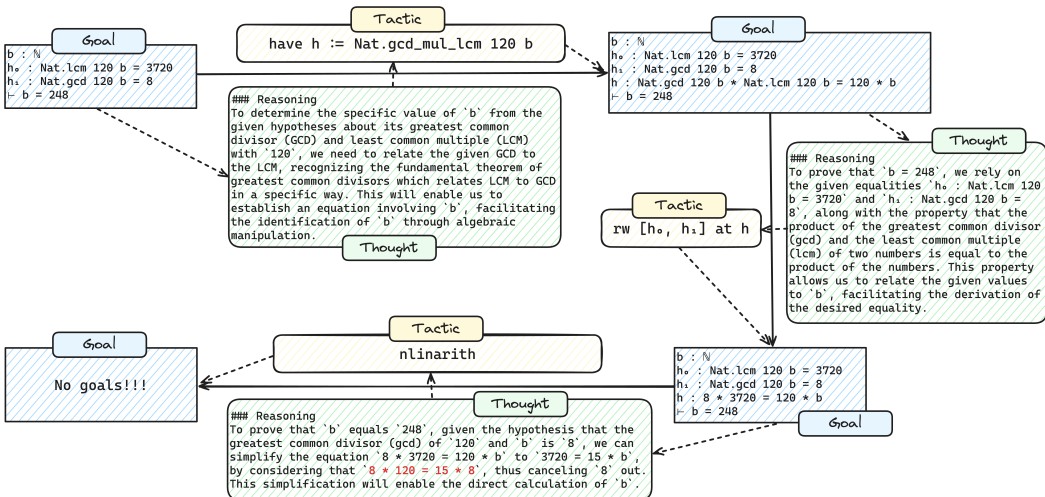

Figure 2: **An example of Lean proof and thoughts generated by Lean-STaR**. Note that there is a calculation error in the thought (in red), but this does not affect the correctness of the proof because the calculation task is actually completed by the interactive theorem prover (i.e., Lean's `nlinarith`) instead of the language model. This shows a benefit of combining neural and symbolic systems.

### 3.2.1 DIRECT TACTIC PREDICTION

We define the theorem-proving problem as a *Markov Decision Process* (MDP) $(\mathcal{S}, \mathcal{A}, P_a, R_a)$ where proof states serve as states in MDP and tactics serve as actions. From this perspective, a proof is a trajectory $(s_1, a_1, r_1), (s_2, a_2, r_2), \cdots$ of states $s_i$, tactics $a_i$, and rewards $r_i \in \mathbb{R}$, and the ITP (e.g., Lean) provides each new state $s_{i+1}$.

In the typical setting (Polu & Sutskever, 2020), proving a theorem consists of providing a proof state $s$ to the language model and then generating a tactic from the language model $M$, i.e., $\pi_M(a|s)$. The language model can be fine-tuned for this task using a dataset of (proof state, next-tactic) pairs from successful proof trajectories, i.e. $D = \{(s^i, a^i) : i = 1, \cdots, M\}$, where final states have a reward of 1. We refer to a language model fine-tuned on such a dataset as a *supervised fine-tuning (SFT)* model.

### 3.2.2 THOUGHT-AUGMENTED TACTIC PREDICTION

Existing approaches typically train only on formal states and tactics (Polu & Sutskever, 2020). We hypothesize that incorporating a latent *thought* can improve a model's ability to predict the next tactic. Formally, we introduce a hidden "thought" variable $t_i$ prior to each tactic, and then extend the model to the form $\pi_M(a_i, t_i|s_i) = \pi_M(a_i|t_i, s_i)\pi_M(t_i|s_i)$. In thought-augmented tactic prediction, the distribution over the next tactic can then be expressed as:

$$\pi_M(a_i|s_i) = \sum_{t_i} \pi_M(a_i|t_i, s_i)\pi_M(t_i|s_i).$$

The key challenge is obtaining (state, thought, tactic) pairs for training a model. To this end, we introduce **retrospective rationale generation**. Our motivating observation is that the distribution of natural language thoughts in theorem-proving $\pi_M(t_i|s_i)$ is scarce in the pre-training corpus of large language models. In turn, we find that even the most powerful GPT-4 model does not perform well in generating the correct rationale through few-shot prompting (Brown et al., 2020). To develop a language model capable of generating thoughts and tactics $a_i, t_i|s_i$, we need an entirely new dataset $D_T = \{(s^i, t^i, a^i) : i = 1, \cdots, N\}$. However, in Lean, we only have a dataset of $D_S = \{(s^i, a^i) : i = 1, \cdots, N\}$ where $(s^i, a^i)$ is one step in some successful proof trajectories. Given a powerful large language model $G$, which we refer to as the oracle model[2], we give the oracle

---

[2]For instance, in our experiments we use the best available large language model, GPT-4.

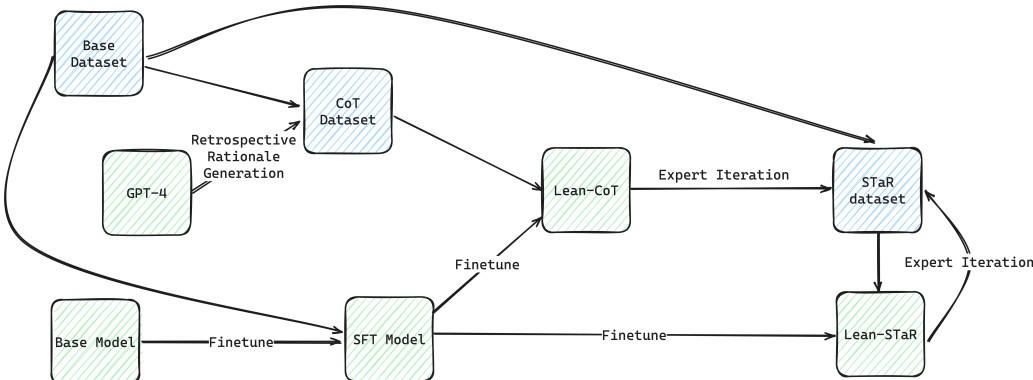

Figure 3: **The diagram of our pipeline.** (1) Produce CoT dataset through GPT-4. (2) Fine-tune the SFT model with the CoT dataset to obtain Lean-CoT. (3) Use expert iteration to generate the STaR dataset through the model in the last iteration (Lean-CoT in the first iteration) and then fine-tune Lean-CoT on the updated STaR dataset to obtain the model in the next iteration. We continue performing this step until a stopping condition is met (e.g., a fixed number of iterations).

model the ground-truth tactic $a_i$ and let the oracle model produce the thought $t_i$ given the current state $s_i$ and ground-truth tactic $a_i$. This helps improve the pass rate and produce thought-augmented data more efficiently. Our few-shot prompt is provided in Appendix G. The design principle of the prompt is to prevent the oracle model from generating hindsight-like thoughts.

We randomly select $M$ pairs $(s^i, a^i) \in D_S$ . Then the oracle model is used to produce a thought $t^i$ for each pair $(s^i, a^i)$ to create a new dataset $D_T\{(s^i, t^i, a^i) : i = 1, \cdots, M\}$. With this retrospectively annotated dataset by the oracle model $D_T$, we obtained our first thought-augmented tactic prediction model, Lean-CoT, by fine-tuning from the SFT model.

### 3.2.3 BOOTSTRAPPING THOUGHT-AUGMENTED THEOREM PROVING

We propose to apply expert iteration to further improve the performance of Lean-CoT. Specifically, we start from the initial Lean-CoT model $M_0$ and the initial dataset $D = \{s^i : i = 1, \cdots, M\}$, which consists of all initial states $s^i$ of the theorems to be proved. In iteration 1, we use model $M$ to sample $K$ times per problem. Each time the model will produce a proof trajectory $[(s_0, t_0, a_0), (s_1, t_1, a_1), \cdots, (s_n, t_n, a_n)]$. Then we create a new dataset $D_1$ by filtering the generated trajectories to include only the successful ones. De-duplication is then applied to the collected trajectories. Now, we can further fine-tune the SFT model $M$ on dataset $D_T \cup D_1$ to produce Lean-STaR model $M_1$. Then we can use $M_1$ as initial model to produce dataset $D_2$ and further fine-tune to obtain model $M_2$ in the next iteration.

This method can be seen as an offline RL method (Singh et al., 2023) in the theorem proving MDP. In this MDP, the cumulative reward $R\left((s_0, t_0, a_0), (s_1, t_1, a_1), \cdots, (s_n, t_n, a_n)\right) = 1$ if and only if the proof trajectory is successful. The total expected reward is

$$J(M, D) = \sum_i \mathbb{E}_{(s_0, t_0, a_0), \cdots, (s_n, t_n, a_n) \sim \pi_M(\cdot | s^i)} R\left((s_0, t_0, a_0), \cdots, (s_n, t_n, a_n)\right),$$

and Lean-STaR's expert iteration can be seen as optimizing this reward (Singh et al., 2023).

### 3.3 EVALUATION

**Setup.** We evaluate the model on formal theorem proving – given a theorem statement, produce a theorem that is correct according to the formal system. This requires an algorithm for producing a full proof by interacting with Lean. As a new form of theorem-proving system, it is unclear what the best strategy is when we have informal thoughts. Our preliminary experiments indicate that best-first search with beam search does not work well for the thoughts in the natural language format. Thus we describe the traditional strategy (best-first search), and our new approach based on sampling.

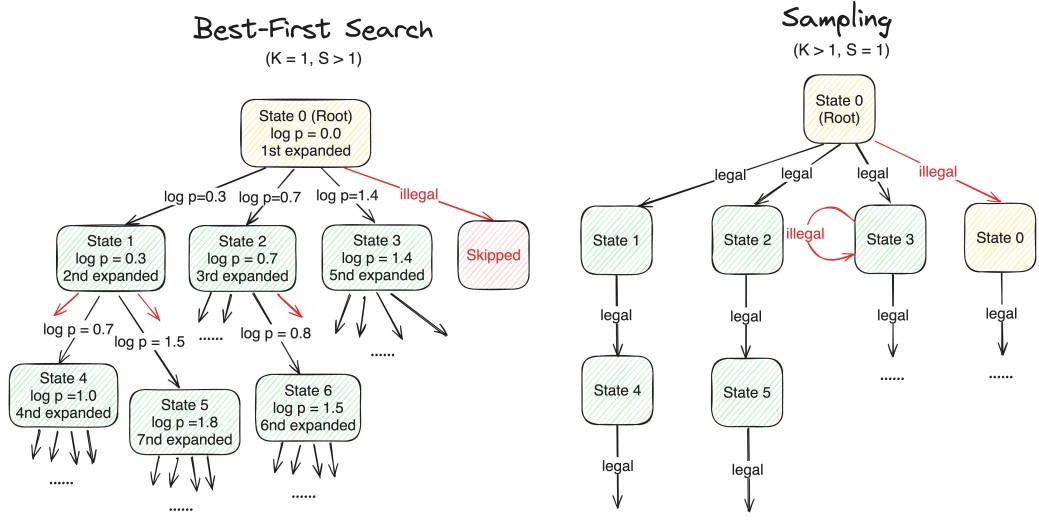

Figure 4: **The visualization of Best-first Search ($K = 1$) and Sampling ($S = 1$).** Search method maintains a search tree and explores $S$ tactics on each expanded node. Sampling method explores $K$ tactic trajectories from the root and ignores illegal tactics in the trajectories.

**Best-First Search.** The most popular method to evaluate the theorem proving ability of a language model $M$ is to use best-first search like GPT-f (Polu & Sutskever, 2020; Yang et al., 2023; Azerbayev et al., 2023b; Welleck & Saha, 2023). In best-first search, we keep all unexpanded states $s_i$. Each time, we expand the "best" state $s_i$ and use the language model to sample $S$ next tactics $a_{i,1\cdots S}$ for the current state $s_i$. For each legal tactic $a_{i,j}$, a new state can be obtained by applying tactic $a_{i,j}$ on state $s_i$. Following standard practice (Polu & Sutskever, 2020; Yang et al., 2023; Welleck & Saha, 2023), we assume the state with maximum negative log-probabilities is the "best"s. Specifically, we select state $s_i$ with maximum $\sum_{j=0}^{i-1} - \log p(a_j, s_j)$, where $(s_0, a_0), \cdots, (s_{i-1}, a_{i-1})$ is the proof trajectory before state $s_i$ and $\log p(a_j, s_j)$ is the average log probability of each generated token. We expand up to $N$ states and we get a successful proof search when we reach any proof state with no goals. Then, we can attempt the search $K$ times to obtain a pass rate $pass@K$. However, we found that the best-first search method performed poorly in the Lean-CoT and Lean-STaR models, as detailed in the Appendix E. We attribute this to using average log probabilities, which may not be a reliable quality indicator when the thought sequence $t_j$ is generated.

**Sampling.** Motivated by these issues with applying best-first search to thought-augmented proving, we develop a new method based on sampling trajectories in parallel. Specifically, our method samples $K$ times in parallel for each problem, each time generating at most $N$ tactics. Also, illegal sampled tactics will be ignored during sampling. Specifically, in a sample, suppose our current state is $s_i$, the proof trajectory before $s_i$ is $(s_0, a_0), \cdots, (s_{i-1}, a_{i-1})$ and the sampled tactic is $a_i$. If $a_i$ is a legal tactic, $(s_i, a_i)$ will be added to the proof trajectory and we will reach a new state obtained by applying tactic $a_{i,j}$ on state $s_i$. Otherwise, we ignore this $a_i$ and use language model $M$ to sample a new tactic given state $s_i$. We limit the number of times a tactic can be generated by language model $M$ to a total of $N$ per time in $K$ sampling times. The sampling method is roughly equivalent to the search with $S = 1$, except that the sampling ignores illegal tactics. We assume that in the sampling method we have $S = 1$. In this setting, evaluating our sampling method and best-first search with equal $S \times K$ took approximately the same amount of GPU time. This sampling method can easily accommodate hidden variable "thoughts" $t_j$. Figure 4 compares best-first search and our sampling method.

Table 1: **Pass rates on the minif2f-test and Leandojo dataset with Lean.** This table shows the pass rates of previous works and our work. $S$ is the number of tactics attempted at each expanded node (assumed to be 1 in sampling) and $K$ is the total number of search or sampling attempts per problem. In sampling we use temperature 0.7, and in search we use beam search when generating the next tactic. We use a random subset of Leandojo4-v9-test (novel premises) with a size of 320 as test set of leandojo. Note that we sample 32 examples twice when $K = 64$ in sampling.

| APPROACH | DECODING | $N$ | $K$ | $S$ | MINIF2F | LEANDOJO |
|---|---|---|---|---|---|---|
| GPT-3.5 ACHIAM ET AL. (2023) (FEW-SHOT) | SAMPLING | 50 | 1 | 1 | 2.8% | - |
| GPT-4 ACHIAM ET AL. (2023) (FEW-SHOT) | SAMPLING | 50 | 1 | 1 | 11.9% | - |
| TRANSFORMER POLU ET AL. (2022) (W/O RL) | SEARCH | 512 | 1 | 8 | 24.6% | - |
| LLEMMA-34B AZERBAYEV ET AL. (2023B) | SEARCH | 50 | 1 | 32 | 25.8% | - |
| LLEMMA-7B AZERBAYEV ET AL. (2023B) | SEARCH | 50 | 1 | 32 | 26.2% | - |
| REPROVER YANG ET AL. (2023) | SEARCH | 50 | 1 | 64 | 26.5% | - |
| TRANSFORMER POLU ET AL. (2022) (W/ RL) | SEARCH | 512 | 1 | 8 | 29.6% | - |
| INTERNLM2-34B YING ET AL. (2024) | SEARCH | 50 | 1 | 32 | 29.5% | - |
| COPRA (WITH GPT-4) THAKUR ET AL. (2023) | CUSTOMIZED | - | 60 | 1 | 29.9% | - |
| COPRA (WITH GPT-4) THAKUR ET AL. (2023) | CUSTOMIZED | - | 100 | 1 | 30.7% | - |
| INTERNLM2-7B YING ET AL. (2024) | SAMPLING | 50 | 32 | 1 | 28.7% | 29.7% |
| INTERNLM2-7B YING ET AL. (2024) | SEARCH | 50 | 1 | 32 | 30.3% | - |
| SFT (INTERNLM2-7B) | SAMPLING | 50 | 32 | 1 | 29.5% | 30.6% |
| SFT (INTERNLM2-7B) | SEARCH | 50 | 1 | 32 | 30.7% | - |
| **LEAN-COT** (INTERNLM2-7B) | SAMPLING | 50 | 32 | 1 | 32.8% | 35.6% |
| **LEAN-STAR (ITER-1)** (INTERNLM2-7B) | SAMPLING | 50 | 32 | 1 | 34.0% | 38.4% |
| **LEAN-STAR (ITER-2)** (INTERNLM2-7B) | SAMPLING | 50 | 32 | 1 | **34.8**% | **39.4**% |
| **LEAN-STAR (ITER-2)** (INTERNLM2-7B) | SAMPLING | 50 | 64 | 1 | **36.1**% | - |

## 4 EXPERIMENTS

We instantiate Lean-STaR using the best available open language model pre-trained on the Lean corpus (InternLM2-Math-base-7b (Ying et al., 2024)), and follow standard practice in using Lean's Mathlib as the underlying training set (via the Lean Dojo dataset (Yang et al., 2023)). We generate an initial set of thoughts for Mathlib using GPT-4, perform two rounds of expert iteration, then evaluate the model on miniF2F (Zheng et al., 2021) and leandojo (Yang et al., 2023), the de-facto standard benchmark for evaluating language-model based theorem provers. Our experimental results show that both retrospective rationale generation and expert iteration significantly improve the theorem-proving capabilities of language models in this setting. We describe our setup and findings in detail below.

### 4.1 EXPERIMENTAL SETUP

We use *LeanDojo Benchmark 4 v9* as the supervised fine-tuning (SFT) dataset containing $231,240$ data examples. We fine-tune for 1 epoch to obtain the SFT model. For the learning rate, we use a warmup in the first $20\%$ steps from 0 to $2 \times 10^{-5}$, followed by a cosine schedule decaying to zero.

We randomly select $17,256$ different successful proof trajectories from *LeanDojo Benchmark 4 dataset* (Yang et al., 2023), and use GPT-4-0125 (OpenAI, 2023) to annotate $52,438$ thoughts from those proof trajectories. We filtered out all proof steps $(s^i, a^i)$ for which $a^i$ contains the newline symbol "\n" before annotating. We perform two iterations of expert iteration, and provide the details in Appendix A.1 due to space.

We evaluate our method on the *MiniF2F* benchmark (Zheng et al., 2021). We use a similar evaluation setting as previous works (Yang et al., 2023; Welleck & Saha, 2023; Ying et al., 2024), but use our

Table 2: **Pass rates about InternLM2-Plus-7B on the minif2f-test dataset with Lean.** This table shows the pass rates of previous works and our work. The evaluation setting is the same as Table 1.

| APPROACH | DECODING | $N$ | $K$ | $S$ | PASS RATE |
|---|---|---|---|---|---|
| INTERNLM2-PLUS-7B (YING ET AL., 2024) (FROM PAPER) | SEARCH | 1000 | 1 | 32 | 43.4% |
| INTERNLM2-PLUS-7B (YING ET AL., 2024) (REPRODUCED) | SEARCH | 1000 | 1 | 32 | 42.6% |
| INTERNLM2-PLUS-7B (YING ET AL., 2024) | SAMPLING | 50 | 32 | 1 | 40.9% |
| SFT (INTERNLM2-PLUS-7B) (YING ET AL., 2024) | SAMPLING | 50 | 32 | 1 | 41.3% |
| **LEAN-COT** (INTERNLM2-PLUS-7B) | SAMPLING | 50 | 32 | 1 | 43.4% |
| **LEAN-STAR (ITER-1)** (INTERNLM2-PLUS-7B) | SAMPLING | 50 | 32 | 1 | 45.4% |
| INTERNLM2-PLUS-7B (YING ET AL., 2024) | SAMPLING | 50 | 64 | 1 | 42.2% |
| SFT (INTERNLM2-PLUS-7B) (YING ET AL., 2024) | SAMPLING | 50 | 64 | 1 | 43.4% |
| **LEAN-COT** (INTERNLM2-PLUS-7B) | SAMPLING | 50 | 64 | 1 | 45.5% |
| **LEAN-STAR (ITER-1)** (INTERNLM2-PLUS-7B) | SAMPLING | 50 | 64 | 1 | **46.3**% |

sampling method instead of best-first search for the evaluation of our thought-augmented theorem proving model as discussed in (§3.3). We choose these settings to resemble the inference budget used in our baselines, which follow previous work (Welleck & Saha, 2023; Azerbayev et al., 2023b; Ying et al., 2024). Namely, for best-first search baselines we use beam search to generate the next tactic with $S = 32, K = 1$ (Welleck & Saha, 2023; Azerbayev et al., 2023b; Ying et al., 2024). We do not compare with methods designed for other formal languages such as Jiang et al. (2022); Xin et al. (2023) since language differences greatly influence the pass rate due to the different tactics and automation. We also do not compare with Lample et al. (2022) since they only report $S = 32, K = 64$ on best-first search, which is approximately equivalent to $S = 1, K = 512$ for the sampling method, which is too computationally expensive for us.

### 4.2 MAIN RESULTS

Our main results are reported in Table 1. Lean-STaR gives a significant improvement over the base model. For instance, with a similar inference budget, Lean-STaR achieves 34.8% versus 30.3% in InternLM2 (Ying et al., 2024) using best-first search and 30.7% in COPRA (Thakur et al., 2023) using GPT-4. With a larger compute budget, Lean-STaR's performance improves further to 36.1%.

**Thought augmentation improves theorem proving.** The first phase of Lean-STaR trains a model to interleave thoughts and tactics, by fine-tuning on a synthesized dataset of thought-augmented examples. The fine-tuned model from this phase, denoted LEAN-COT in Table 1, achieves a pass rate of 32.8%, which is higher than the model prior to this phase, denoted SFT (29.5%). We conclude that the first phase of Lean-STaR can improve the theorem proving ability of a language model, even one that is already specialized for generating tactics in Lean such as the SFT model.

**Bootstrapping improves thought-augmented theorem proving.** The second phase of Lean-STaR consists of generating new thoughts and tactics with the current language model, saving those that result in correct proofs, and training on the union of the initial thought-augmented dataset and the saved examples (i.e., expert iteration (Polu et al., 2022; Zelikman et al., 2022; Singh et al., 2023)). Refer to Appendix A.1 for details.

We perform two iterations of expert iteration, and present the results in Table 1, denoted LEAN-STAR. Each iteration improves the model's theorem proving performance, from 32.8% (the initial model) to 34% (LEAN-STAR after iteration 1) to 34.8% (LEAN-STAR after iteration 2). Furthermore, we find that the model is amenable to further improvement via additional sampling, achieving 36.1% by doubling the sampling budget. We conclude that Lean-STaR's second phase can further improve a model's ability to generate thoughts and tactics that lead to correct proofs. We include three qualitative examples in the Appendix, which show the model interleaving thoughts and proof steps.

Table 3: Results for the InternLM2-plus-7b and our Lean-CoT, Lean-STaR, and expert iteration without CoT. We use sampling with $N = 50, K = 32, \& T = 0.7$.

| APPROACH | *Pass@32* OF INTERNLM-BASE | *Pass@32* OF INTERNLM-PLUS |
|---|---|---|
| FEW-SHOT | 28.7% | 40.9% |
| SFT | 29.5%(+0.8%) | 41.3%(+0.4%) |
| LEAN-COT | 32.8%(+**3.3**%) | 43.4%(+**2.1**%) |
| LEAN-STAR | 34.0%(+1.2%) | 45.5%(+**2.1**%) |
| EXPERT ITERATION (SFT) | 30.7%(+1.2%) | 43.0%(+1.7%) |

### 4.3 EXPERIMENTS WITH STRONGER BASE MODEL AND MORE DATA

We instantiate Lean-STaR using a stronger language model (InternLM2-Math-plus-7b (Ying et al., 2024)), which was released after the experiment above. We follow a similar setup to the previous experiment.

In this experiment, we used $140,000$ thoughts annotated by GPT-4o (OpenAI, 2023) to fine-tune a model ("Lean-CoT"). Then we performed only one iteration of expert iteration and collected about $60,000$ (proof state, thoughts, next-tactic) pairs in data, named "STaR dataset" $D_1$. We further fine-tuned the Lean-CoT model on dataset $D_1$ to get the Lean-STaR model.

Our new results are reported in Table 2. We can see that Lean-STaR still gives a significant improvement over the baseline. For instance, Lean-STaR achieves $45.4\%$ versus $40.9\%$ in InternLM-plus using sampling with a similar inference budget and $43.4\%$ using best-first search with more inference budget reported in (Ying et al., 2024). This results show that both retrospective rationale generation and expert iteration can improve the theorem-proving capabilities on a stronger base model.

### 4.4 EXPERIMENTS ON EXPERT ITERATION WITHOUT COT

Table 3 shows the result of expert iteration without CoT (i.e., using (state, tactic) pairs only) as well as the result of Lean-CoT and Lean-STaR. Expert iteration alone achieves 43.0%, which is less than Lean-STaR (45.4%) in InternLM-plus and achieves 30.7% verus 34.0% in InternLM-base. This shows that Lean-STaR's performance gains do not only come from the use of expert iteration.

## 5 CONCLUSION & LIMITATIONS

In this paper, we presented Lean-STaR, a novel approach that significantly enhances the theorem-proving capabilities of language models in formal mathematics by integrating Chain-of-Thought (CoT) rationales into each proof step. Our method begins with generating synthetic rationales using ground-truth tactics retrospectively, followed by fine-tuning the language model to generate these rationales and predict subsequent tactics, resulting in the Lean-CoT model. We further improved this model using expert iteration, fine-tuning it on correct proofs it samples and verifies using the Lean solver. Our contributions include the introduction of the first thought-augmented theorem proving dataset, demonstrating that expert iteration can further improve performance, and achieving new results on the miniF2F-test benchmark, increasing the pass rate from $30.3\%$ to $36.1\%$. These advancements are not only about improving the accuracy of automated theorem proving, but also offer a scalable and efficient framework for advancing human understanding of mathematics, which may lead to significant impacts in education, scientific discovery, and program verification (Carter & Monks, 2013; Kang et al., 2020; Szegedy, 2020; Avigad, 2023; First, 2023; of Sciences, 2023).

The primary limitation of our method is that its performance may be constrained by issues of computational scalability. Both Lean-CoT and Lean-STaR have been fine-tuned on a dataset that is not very large. Additionally, the use of GPT-4 to generate synthetic data may incur a significant

cost and possibly introduce biases. Also, expert iteration could face a bottleneck due to CPU and IO limitations, which might slow down the process due to a sluggish speed of Lean ITP.

## ACKNOWLEDGMENTS

We thank the anonymous reviewers and area chair for their helpful comments. Zhiqing Sun acknowledges the support of the Google PhD Fellowship. Sean Welleck thanks NSF SCALE (NSF DMS 2134012) and Convergent Research.

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

## A ADDITIONAL EXPERIMENT SETUP

### A.1 LEAN-STAR EXPERT ITERATION

The second phase of Lean-STaR consists of generating new thoughts and tactics with the current language model, saving those that result in correct proofs, and training on the union of the initial thought-augmented dataset and the saved examples (i.e., expert iteration (Polu et al., 2022; Zelikman et al., 2022; Singh et al., 2023)). We perform two iterations of expert iteration, and provide details on our specific experimental setup below.

In each iteration we use sampling on the *LeanDojo Benchmark 4* dataset, and save the (state, thought, tactic) examples that are part of successful proofs. For each problem, we sample $K = 32$ times in parallel with temperature $T = 1.0$, and limit the number of times a tactic can be generated to a total of $N = 5$ per problem. Also, sampling is limited to 1 minute per problem. In this setup, each problem needs on average about $0.5$ A100 minutes. We collect successfully sampled trajectories to produce a "STaR dataset" $D_1$, and up to 3 proof trajectories were collected for each problem. We collected $32,231$ different (proof state, thoughts, next-tactic) pairs in successful proof trajectories during expert iteration, which takes about 4 days with $8 \times A100$ GPUs. Then, we further fine-tune SFT model for 1 epoch on the combination of GPT-4 annotated reasoning data and expert iteration data $D_T \cup D_1$ to get the Lean-STaR model. We use the same learning rate setup that was used for the SFT model. In the second iteration, we generate a dataset $D_2$ in a similar fashion. Then, we chose to further fine-tune model from iteration 1, $M_1$, on the generated dataset $D_2$ (roughly 19k pairs).

The setup of experiment about InternLM2-plus is slightly different. The details are shown in Section 4.3 and Appendix F.

## B STATISTICS FOR OUR METHODS AS WELL AS THE BASELINES

Table 4: Statistics for the baselines and our Lean-CoT, Lean-STaR on *MiniF2F* dataset. We use sampling method with hyperparameters $N = 50$ & $K = 32$ & $T = 0.7$.

| APPROACH | # (CONTINUAL) TRAINING DATA | Pass@32 | |
|---|---|---|---|
| INTERNLM2-MATH-7B (FEW-SHOT) | - | 28.7% | - |
| SFT | $231,240$ | 29.5% | +0.8% |
| **LEAN-COT** | $52,438$ | 32.8% | **+3.3%** |
| **LEAN-STAR (ITER-1)** | $32,231$ | 34.0% | +1.2% |
| **LEAN-STAR (ITER-2)** | $19,324$ | **34.8%** | +0.8% |

## C   DATA LEAKAGE

A risk of using GPT-4 for generating thought annotations is data leakage. Sine miniF2F has been used as a dataset for formal theorem proving for a while, it is possible that miniF2F dataset is already seen by GPT-4 or internLM during pre-training process. However, there are several reasons to believe that data leakage is not likely to be true.

First, our experimental setting (fine-tuning on Mathlib, evaluating on miniF2F) follows a widely used experimental setup in benchmark evaluations in neural theorem proving and InternLM was also evaluated on miniF2F. Therefore, we believe that InternLM has not been exposed to miniF2F.

Also, we observed that most of the proofs generated by Lean-STaR are completely different from the manually written proofs in the miniF2F test dataset. Table 5 shows the analysis of proof generated by Lean-STaR on miniF2F test dataset for Lean. From Table 5 we can see that almost all proofs generated by LeanSTaR are different from the proofs mentioned in the miniF2F. If we set aside straightforward simple cases, there is only one proof generated by LeanSTaR is the same as the proofs mentioned in the miniF2F.

Table 5: Analysis of proof generated by Lean-STaR on miniF2F test dataset for Lean. Similar to the analysis in Thakur et al. (2024).

| | Proofs found in miniF2F-test | | | | | | Proofs NOT in miniF2F | Total |
|---|---|---|---|---|---|---|---|---|
| | Single-Tactic Simple Proofs | | | Two-Tactic Proofs | Longer OR Complex Proofs | Total | | |
| Tactics Used | linarith | norm_num | nlinarith | two tactics | > 2 tactics OR 1 tactic multi-args | | sorry | |
| Proof Count | 11 | 12 | 2 | 18 | 39 | 82 | 162 | 244 |
| Exact Match Successful Proof Count | 2 | 4 | 0 | 0 | 1 | 7 | 0 | 7 |
| 1st Tactic Match Successful Proof Count | 2 | 4 | 0 | 0 | 1 | 7 | 0 | 7 |
| Distinct Successful Proof Count | 5 | 7 | 2 | 14 | 19 | 47 | 43 | 90 / 112 (80.36%) |
| Distinct Successful Proof Count ex Single-Tactic | - | - | - | 6 | 12 | 18 | 20 | 38 |
| All Successful Proof Count | 11 | 12 | 2 | 17 | 22 | 64 | 48 | 112 |

## D   AN EXAMPLE AND EXPLANATION OF A FORMAL PROOF IN LEAN

An example of a formal proof in Lean with its visualization is shown in Figure 5, taken from (Lample et al., 2022). In the proof, the tactic `induction k` is is applied to the initial state $(n \leq m \Rightarrow n + k \leq m + k)$ and the ITP converts the current state to subgoals `case 0` $\wedge$ `case ih`: $n \leq m \wedge n + k \leq m + k \Rightarrow n + (k+1) \leq m + (k+1)$. The `case 0:` $n \leq m$ is our hypothesis $h_0$ so it can be proven by `case 0:exact` $h_0$ tactic. Then, we rewrite the `case ih` through the `nat.succ_le_succ_iff` which is a theorem in Lean library means $n \leq m \Leftrightarrow n + 1 \leq m + 1$.

After tactics `case 0:exact` $h_0$ and `case ih:rw nat.succ_le_succ_iff`, the goal state is converted to $n + k \leq m + k$ which is the hypothesis introduced by induction. Therefore, we can complete this proof using tactic `exact k_ih`.

```
theorem add_le_add_right (m n k : ℕ) (h₀ : n ≤ m)
    : n + k ≤ m + k :=
    induction k with
    | zero =>
        exact h₀
    | succ k ih =>
        rw Nat.succ_le_succ_iff
        exact ih
```

Figure 5: **A example proof and its visualization of** $n \leq m \Rightarrow n + k \leq m + k$ **in Lean, taken from (Lample et al., 2022).** The `induction` tactic reduces the initial statement to two subgoals. Then tactics `case 0:exact` $h_0$ and `case ih:rw nat.succ_le_succ_iff`, `case ih:exact k_ih` can be applied in turn to complete the proof.

Table 6: Counts of problems successfully proved in *minif2f-test* benchmark, split by type and difficulty. The methods use sampling with $N = 50, K = 32$. Thought-augmented methods improve performance on all categories, while Lean-STaR significantly improves Number Theory performance.

| | TOTAL | | TEST SET SIZE | INTERNLM2-7B | SFT | LEAN-CoT | LEAN-STAR (ITER-2) |
|---|---|---|---|---|---|---|---|
| | IMO | | 20 | 0 | 0 | 0 | 0 |
| | AIME | | 15 | 2 | 1 | 2 | **3** |
| | AMC | | 45 | 3 | 3 | **7** | 5 |
| | | LEVEL 5 | 14 | 1 | 2 | **3** | **3** |
| | | LEVEL 4 | 14 | **7** | **7** | **7** | **7** |
| | ALGEBRA | LEVEL 3 | 14 | 9 | 9 | **11** | **11** |
| | | LEVEL 2 | 14 | 10 | 10 | 9 | **11** |
| MATH | | LEVEL 1 | 14 | 9 | **10** | **10** | **10** |
| | | LEVEL 5 | 16 | 6 | 6 | 6 | **7** |
| | | LEVEL 4 | 11 | **5** | **5** | 4 | **5** |
| | NUMBER THEORY | LEVEL 3 | 11 | 4 | 5 | 5 | **6** |
| | | LEVEL 2 | 11 | **6** | 5 | 5 | **6** |
| | | LEVEL 1 | 11 | 8 | 8 | **9** | **9** |
| | ALGEBRA | | 18 | 0 | **1** | **1** | **1** |
| CUSTOM | NUMBER THEORY | | 8 | 0 | 0 | 0 | 0 |
| | INDUCTION | | 8 | 0 | 0 | **1** | **1** |

sectionPerformance Analysis by Types and Difficulties Tasks in *minif2f-test* are manually formalized from Olympiad type problems, drawn from multiple sources including AIME, AMC, IMO problems, and problems from the MATH dataset (Hendrycks et al., 2021). These problems can have different levels of difficulty and types. Table 6 reports the number of problems successfully proved, partitioned by type and difficulty. We see that Lean-CoT improves performance in solving difficult problems on all categories, especially those from mathematics competitions. On top of these improvements, Lean-STaR's improvements come mainly in Number Theory.

### D.1 PERFORMANCE ANALYSIS BY TYPES AND DIFFICULTIES USING INTERNLM2-PLUS-7B

Table 7 reports the number of problems successfully proved, partitioned by type and difficulty using InternLM2-plus. We see that Lean-CoT improves performance mainly in Number Theory and Lean-STaR improves performance in solving difficult problems on all categories, which is the opposite of the performance of the InternLM2-base.

### E COMPARISON BETWEEN SEARCH METHOD AND SAMPLING METHOD

Table 7: Counts of problems successfully proved in *minif2f-test* benchmark using InternLM2-plus-7b, split by type and difficulty. The methods use sampling with $N = 50, K = 32$.

| TOTAL | | | TEST SET SIZE | INTERNLM2-PLUS-7B | LEAN-CoT | LEAN-STAR (ITER-1) |
|---|---|---|---|---|---|---|
| IMO | | | 20 | 0 | 0 | 0 |
| AIME | | | 15 | 3 | 3 | **4** |
| AMC | | | 45 | 9 | 9 | **10** |
| MATH | ALGEBRA | LEVEL 5 | 14 | **6** | **6** | 6 |
| | | LEVEL 4 | 14 | **9** | **9** | 9 |
| | | LEVEL 3 | 14 | 11 | **13** | 13 |
| | | LEVEL 2 | 14 | **11** | **11** | 11 |
| | | LEVEL 1 | 14 | **10** | **10** | 10 |
| | NUMBER THEORY | LEVEL 5 | 16 | **7** | **7** | 7 |
| | | LEVEL 4 | 11 | 6 | **8** | 8 |
| | | LEVEL 3 | 11 | 6 | 7 | **9** |
| | | LEVEL 2 | 11 | 7 | **9** | 9 |
| | | LEVEL 1 | 11 | **10** | **10** | 10 |
| CUSTOM | ALGEBRA | | 18 | **4** | 3 | 4 |
| | NUMBER THEORY | | 8 | 0 | 0 | 0 |
| | INDUCTION | | 8 | **1** | **1** | 1 |

Table 8: **Comparison between search method and sampling method.** We use sampling method with hyperparameters $N = 50 \ \& \ S = 1 \ \& \ K = 32$ and BFS method with $N = 50 \ \& \ S = 32 \ \& \ K = 1$. All sampling decoding in the paper uses a temperature of 0.7. We use BFS to denotes Best-First Search.

| APPROACH | BFS (SAMPLING) | BFS (BEAM SEARCH) | SAMPLING |
|---|---|---|---|
| TACTIC PREDICTION IN PROVING | BFS | BFS | SAMPLING |
| TOKEN DECODING IN TACTICS | SAMPLING | BEAM-SEARCH | SAMPLING |
| INTERNLM2-7B (FEW-SHOT) | 29.1% | 30.3% | 28.7% |
| SFT | 29.9% | 30.7% | 29.5% |
| LEAN-CoT | 27.0% | 25.4% | 32.8% |
| LEAN-STAR (ITER-1) | 29.1% | 26.2% | 34.0% |
| LEAN-STAR (ITER-2) | 29.5% | 26.2% | 34.8% |

## F PERFORMANCE DIFFERENCE OF JOINT TRAINING AND CONTINUE TRAINING

As shown in Table 9, the joint training method performs better using InternLM2-base but training method performs much better using InternLM2-plus. It seems that there are no difference between these two methods. Therefore, this performance can be depend on the quantity of data or the model. (We use much more data when using InternLM2-plus and the quantity of "STaR data" is relatively small.)

### F.1 SEARCH AND SAMPLING BUDGET

Table 10 reports the trends of the pass rate against the search size or sampling budget $S \times K$. We find that Lean-STaR benefits more as $K$ increases, especially when $K$ is relatively large. The result suggests that additional sampling with thoughts improves performance, while additional sampling without thoughts may saturate. We believe this is because thoughts increase the diversity of outputs and contribute to exploration in the theorem proving space. Therefore, Lean-STaR is more scalable

Table 9: Performance difference of joint training and continue training on Lean-STaR. We use sampling method with hyperparameters $N = 50 \ \& \ K = 32 \ \& \ T = 0.7$. In continue training, we further fine-tune the Lean-CoT model on "STaR data" to get Lean-STaR model and in joint training we fine-tune the SFT model on combination of GPT-4 annotated reasoning data and "STaR data".

| APPROACH | INTERNLM2-BASE-7B | INTERNLM2-PLUS-7B |
|---|---|---|
| **LEAN-CoT** | 32.8% | 43.4% |
| **LEAN-STAR (ITER-1)** (JOINT TRAINING) | **34.0%** | 43.9% |
| **LEAN-STAR (ITER-1)** (CONTINUE TRAINING) | 33.2% | **45.5%** |

Table 10: Performence of SFT-Direct and our Lean-STaR at different search size or sampling times $S \times K$. We fix $N = 50$. We use beam search in search and temperature $T = 0.7$ in sampling when generating the next tactic. We have $K = 1$ in search and $S = 1$ in sampling. Note that we sample 32 examples twice when $K = 64$ in sampling.

| | SFT-DIRECT (SEARCH) | SFT-DIRECT (SAMPLING) | LEAN-STAR (ITER-2) (SAMPLING) |
|---|---|---|---|
| $S \times K = 1$ | 13.5% | 20.9% | 21.7% |
| $S \times K = 2$ | 18.0% (+4.5%) | 22.5% (+1.6%) | 24.6%(+2.9%) |
| $S \times K = 4$ | 23.3% (+5.3%) | 25.0% (+2.5%) | 27.5%(+2.9%) |
| $S \times K = 8$ | 27.5% (+4.2%) | 27.0% (+2.0%) | 30.7% (+3.2%) |
| $S \times K = 16$ | 29.9% (+2.4%) | 28.3% (+1.3%) | 33.6% (+2.9%) |
| $S \times K = 32$ | 30.7% (+0.8%) | 29.5% (+1.2%) | 34.8% (+1.2%) |
| $S \times K = 64$ | 30.7% (+0.0%) | 30.3% (+0.8%) | 36.1% (+1.3%) |

(in terms of inference-time compute), and may be amenable to further improvements with additional iterations of expert iteration.

## G  RETROSPECTIVE RATIONALE GENERATION PROMPT OF GPT-4

```
Please act as a professional mathematician.
Your goal is to accurately prove a math theorem in Lean4.
You are given the first tactic that should be taken to prove the Given
    Theorem.

# Example:

Given Theorem:
```lean4
X : Type u_1
Y : Type u_2
inst✝¹ : TopologicalSpace X
inst✝ : TopologicalSpace Y
f : X →. Y
h : PContinuous f
⊢IsOpen (PFun.preimage f Set.univ)
```
Given Tactic:
```lean4
exact h _ isOpen_univ
```

```
### State
a b n : ℕ
⊢ ¬(n ≠ 0 ∧ a ≠ 0) ↔ n = 0 ∨ a = 0
### Tactic
```lean4
    simp only [not_and_or, ne_eq, not_not]
```
```

```
### Reasoning
To prove the equivalence between the negation of a conjunction and a disjunction, we utilize logical
    equivalences, specifically the negation of a conjunction ¬(`(P ∧ Q)`) being equivalent to the disjunction
    of the negations (`P = 0 ∨ Q = 0`) and simplifications related to negation and inequality.
```

Figure 6: Examples of (input, output) pairs of retrospective rationale generation with GPT-4. The full prompt is given in Appendix G. See §3.2.2 for the detailed explanation.

```
```
Assistant Output:
### PLANNING OF REASONING
Given the above goal and the given the target tactic `exact h _
    isOpen_univ`, I should produce a reasoning that only based on the goal
    , and conclude to the `exact h _ isOpen_univ`, but not in the
    hindsight format.

### REASONING

By directly using the hypothesis `h : PContinuous f`, we state that for
    any open set, its preimage via `f` is also open, and this applies to `
    Set.univ` as well.

### TACTIC

```lean4
exact h _ isOpen_univ
```

To achieve the goal, you have three jobs.
# Write down the planning of reasoning.
# Write down the reasoning that can result in the selection of the Given
    Tactic.
# Write down the Given Tactic.
You have four principles to do this.
# Craft reasoning that is comprehensive yet concise, contained ideally
    within two sentences.
# Begin reasoning with the objective of the Given Theorem, ensuring
    logical progression, rather than constructing it retrospectively from
     the Given Tactic.
# Avoid mentioning the Given Tactic and specific statement names from the
     Given Tactic throughout your reasoning.
# Position the conclusion of the tactic after presenting all reasons,
    ensuring that details regarding the Given Tactic are not introduced
    at the start of your reasoning.
Your output should be strictly in the following format and should not
    contain extra content:

### PLANNING OF REASONING

Given the above goal and the given the target tactic <Given Tactic>, I
    should produce a reasoning that only based on the goal, and conclude
    to the exact <Given Tactic>, but not in the hindsight format.

### REASONING
```

```
<your reasoning to the Given Tactic>

### TACTIC

<Given Tactic>

# Given Theorem:
```lean4
{theorem["state_before"]}
```

# Given Tactic:
```lean4
{theorem["tactic"]}
```
```

## H  EXAMPLES OF GENERATED LEAN PROOFS WITH THOUGHTS

```
theorem aime_1990_p4 (x : ℝ) (h₀ : 0 < x) (h₁ : x ^ 2 - 10 * x - 29 ≠ 0)
    (h₂ : x ^ 2 - 10 * x - 45 ≠ 0) (h₃ : x ^ 2 - 10 * x - 69 ≠ 0)
    (h₄ : 1 / (x ^ 2 - 10 * x - 29) + 1 / (x ^ 2 - 10 * x - 45) - 2 / (x
    ^ 2 - 10 * x - 69) = 0)
    : x = 13 :=
  norm_num at h₁ h₂ h₃ -- Reasoning: To prove `x = 13`, we examine the
  conditions under which the given equation holds, specifically
  focusing on the non-equality of the expressions involving `x ^ 2 - 10
  * x - 29`, `x ^ 2 - 10 * x - 45`, and `x ^ 2 - 10 * x - 69`.
  Simplifying these expressions using properties of real numbers and
  algebra allows us to directly compare them with their respective
  values, ultimately identifying `x = 13` as the unique solution that
  satisfies all constraints.
  field_simp at * -- Reasoning: To prove \(x = 13\), we leverage the
  given equalities involving rational fractions of \(x^2 - 10x - c\)
  for various constants \(c\), where \(0 < x\) and each \(x^2 - 10x -
  c\) is non-zero. By algebraically manipulating these expressions, we
  aim to derive a relationship among \(x\) and these constants,
  specifically focusing on \(13\), under the given non-zero conditions
  for the quadratic expressions.
  ring_nf at h₄ ⊢ -- Reasoning: To establish the equality `x = 13`
  under the given conditions, we need to manipulate the algebraic
  expression given by `h₄` to reveal its dependence on `x`, recognizing
  that the presence of `x` in the expressions may affect the equation's
  truth value. The application of algebraic transformations on both
  sides of the equation will facilitate the identification of `x`'s
  value, taking into consideration the inequality conditions provided
  (`h₁`, `h₂`, `h₃`).
  nlinarith -- Reasoning: To prove \(x = 13\), we must manipulate the
  equation given by \(h₄\) to isolate \(x\), using the algebraic
  properties of equality and the assumption \(h₀\) which states \(0 <
  x\). This requires careful rearrangement and application of
  algebraic identities to simplify and rearrange terms, facilitating
  the direct application of the properties of real numbers to reach
  the desired conclusion.

theorem amc12a_2002_p6 (n : ℕ) (h₀ : 0 < n)
    : ∃ m, m > n ∧ ∃ p, m * p ≤ m + p :=
  use n + 1, by linarith only [h₀] -- Reasoning: To demonstrate the
  existence of such an `m` and `p` that satisfy the given inequalities,
  we leverage the property of natural numbers that ensures an increase
  in `n` from `0` leads to the possibility of finding such `m` and `p`.
  This approach is supported by the fundamental property of natural
  numbers and their arithmetic operations, suggesting that for a given
  `n` greater than `0`, there exists a successor `m` and a number `p`
  that satisfy the conditions outlined.
  exact ⟨0, by simp ⟩ -- Reasoning: To demonstrate that there exists
  an \(n \in ℕ\) such that \((n + 1) * p \leq n + 1 + p\) holds for
  some \(p \in ℕ\), we recognize that setting \(p = 0\) satisfies the
  inequality by the properties of zero in arithmetic. This is because
  \(n + 1 > 0\) by assumption, ensuring the left-hand side is
  non-negative and the right-hand side simplifies to \(n + 1\), making
  the inequality trivially true.
```

