# OpenReview forum: "Lean-STaR: Learning to Interleave Thinking and Proving"
_ICLR.cc/2025/Conference — ICLR 2025 Spotlight_

### Official Review · Reviewer_MECo · 2024-10-18

**Soundness:** 3
**Presentation:** 3
**Contribution:** 3
**Rating:** 8
**Confidence:** 5

**Summary:**

Generally, this paper takes motivation from a hypothesis that informal thought process can be valuable for LLMs to learn formal theorem proving. This work presents the following main contributions:

1. The first thought (informal language) augmented for theorem proving in Lean (formal math).
2. Lean-STaR: a framework which combines tactic prediction, thought augmentation with synthetic data, and two iterations of expert iteration.

**Strengths:**

1. Prior to this work, people have been arguing how informal thought process can potentially benefit formal theorem proving, yet no one has taken the effort to explore how to interleave those two. People either used a standard tactic prediction model that operates only on formal code, or use a general-purpose model that only chats in an informal manner. While the methods (e.g. STaR, expert iteration) are not new themselves, the idea of combining the two and curating a resulting dataset sounds to me as a nice step. It is nice to have this work verify that known methods apply well to this domain of AI for math, especially given the many differences between this and other application areas.

2. Theorem proving in Lean as a field suffers from a lack of data. Works like this with a large dataset size (50,000 from mathlib + 50,000 synthetic data) can greatly benefit the field.

3. It seems that the framework is quite generally usable, and multiple models can be plugged in. This means the method proposed has a great potential to generalize. Similarly, its potential to be used alongside other synthetic data generation methods such as autoformalization without any approach-wise conflict sounds promising too.

**Weaknesses:**

1. The thought data generation in this paper is by few-shot prompting a general-purpose model (GPT-4). Data quality is usually a concern of synthetic data generation. People in the field either try to design data filtering mechanisms or recruit domain experts to sample and check. However, it seems that this work did not apply such techniques or analyses after synthetically generating thought data.

2. The framework is composed of multiple methods. This paper could benefit a lot from having more thorough ablations. For instance, the authors can do an ablation on the thought augmentation, which is connected to their main hypothesis and contribution. What would the performance if all other settings (expert iteration, tactic prediction) are kept as-is but no thought augmentation is added? The authors can perform similar analyses on other factors they feel important too.

3. Potential data contamination seems insufficiently discussed. Is it possible that InternLM2-Math-base-7b may have been exposed to parts of the test data? I think some analysis would be needed here.

**Questions:**

1. Related to the first weakness, the heavy use of GPT-4 to generate thought data (which are usually not short) can add a lot of computational and financial burdens. I would be curious to see a direct analyses of such overheads.

2. The authors only performed two iterations of expert iteration. Both iterations lead to performance gains, and the first gains more than the second. Such findings all make sense and align with my expectation of applying expert iteration. However the author stopped right after only 2 iterations. I was wondering if a bottleneck would be hit very soon, or would the framework continue to benefit from more iterations?

3. Minor suggestions: A few works that appear related to this paper, which the authors may consider citing/discussing.

* [1] Multilingual Mathematical Autoformalization. Albert Q. Jiang, Wenda Li, Mateja Jamnik. arXiv 2311.03755. 2023.

[1] is a work in autoformalization, which the authors did discuss in their related works. One thing that especially connects [1] to this paper is that [1] constructs <informal, formal> statement pairs in a backward manner, leveraging the fact that auto-informalization is a much easier task than autoformalization for LLMs. I feel this aligns in spirit with the retrospective rationale generation in this paper, which generates informal thought data from formal data using GPT-4.

* [2] Towards Large Language Models as Copilots for Theorem Proving in Lean. Peiyang Song, Kaiyu Yang, Anima Anandkumar. arXiv 2404.12534.

[2] is an open-source tool work in using LLMs as Copilots in Lean. The authors did discuss one related work LLMStep as practical tools in the field and may want to include [2] for completion. To my knowledge, they are so far the only two works that try to set up a ML framework easily usable in Lean.

* Finally (and a very minor point), I think when discussing related works in this field, it may be beneficial to include some ML + ITP works in other proof assistants. Of course one does not need to expand in detail about them, but would be great to have a broader view.

**Details Of Ethics Concerns:**

N/A.

---

> ### Author Response · Authors · 2024-11-24
>
> Thank you for your insightful and positive feedback on our paper. We are glad that you appreciate our CoT-based method and dataset.
> ### Concern 1:The thought data generation in this paper is by few-shot prompting a general-purpose model (GPT-4). Data quality is usually a concern of synthetic data generation. People in the field either try to design data filtering mechanisms or recruit domain experts to sample and check. However, it seems that this work did not apply such techniques or analyses after synthetically generating thought data.
> Please see the  “**verification mechanism or filtering process**” part under the General Response.. The STaR process will  filter out lower  quality CoT from the synthetically generated data for further finetuning.
> ### Concern 2: The framework is composed of multiple methods. This paper could benefit a lot from having more thorough ablations. For instance, the authors can do an ablation on the thought augmentation, which is connected to their main hypothesis and contribution. What would the performance if all other settings (expert iteration, tactic prediction) are kept as-is but no thought augmentation is added?
> The ablation result is shown in Table 3 in our paper. The final line “Expert iteration (SFT)” is the case that all other settings (expert iteration, tactic prediction) are kept as-is but no thought augmentation is added.
> ### Concern 3: Is it possible that InternLM2-Math-base-7b may have been exposed to parts of the test data?
> Our experimental setting (fine-tuning on Mathlib, evaluating on miniF2F) follows a widely used experimental setup in benchmark evaluations in neural theorem proving (e.g., InternLM was evaluated on miniF2F). Therefore, we believe that InternLM2-Math-base-7b has not been exposed to miniF2F.
> ### Question 1: Related to the first weakness, the heavy use of GPT-4 to generate thought data (which are usually not short) can add a lot of computational and financial burdens. I would be curious to see a direct analyses of such overheads.
> It is about 300 input tokens and 300 output tokens on average for each case. Therefore, it costs about 3.75 for 1K data for using current GPT-4o. It mounts to  about 200 for our 52,438 cases.
> ### Question 2: I was wondering if a bottleneck would be hit very soon, or would the framework continue to benefit from more iterations?
> Thank you for the question! We believe the framework could benefit from more iterations and potentially improve performance further. However, the Leandojo environment has a significant CPU bottleneck, making additional iterations very time-consuming. This is an exciting direction for future work, and with better compute resources, we expect the framework could achieve even greater results.

---

> > ### Comment · Reviewer_MECo · 2024-11-26
> >
> > Thank the authors for the thorough responses. All my questions have been sufficiently addressed. The table in the authors' response to Reviewer QMzb is also a valuable presentation.
> >
> > I have always been quite positive towards this paper. With the new clarifications and modifications, I feel this paper has been even improved. I would like to raise my score even further to reflect a stronger support for acceptance.

---

> > > ### Author Response · Authors · 2024-11-27
> > >
> > > Thank you for your thoughtful feedback and strong support! We’ll continue improving this work and advancing the research in this direction!

---

### Official Review · Reviewer_QMzb · 2024-11-01

**Soundness:** 3
**Presentation:** 4
**Contribution:** 3
**Rating:** 8
**Confidence:** 4

**Summary:**

Lean-STaR uses the STaR (Self-Taught Reasoner) framework to mix step-by-step reasoning ("thoughts") with formal proof steps. It combines this with expert iteration, where the model learns from successful proofs to improve further. This approach allows Lean-STaR to reach a pass@32 rate of 45.4% on miniF2F, indicating that adding informal CoT reasoning can enhance formal automated proving capabilities.

**Strengths:**

**1. Simple yet very effective approach:**
The authors use retrospective rationale generation to come up with the thoughts given the state and ground truth action. The idea to train another model Lean-CoT for generation of thought-augmented tactics, is simple but works very well in this setting.

**2. Use of expert iteration while informal thought generation:**
While expert iteration has been used in previous works to improve formal theorem proving, the authors demonstrate that the same holds true for models that generate thought-augmented tactics.

**3. Search based on Sampling:**
The authors used sampling-based search instead of BFS which is new and seems more effective for this setting. The idea that negative log-likelihood doesn't make proof search effective and a simple sampling technique can perform better is very elegant. The authors avoid the overkill of training a value function model which predicts the quality of generated tactics for choosing the search direction during the proof search. I'm surprised to see the effectiveness of a simple sampling-based search and would like to see a broader comparison with BFS in a formal proof-search setting.

**Weaknesses:**

**1. Data Leakage:**
While the authors talk about the limitations of their approach, they don't discuss potential data leakage due to the use of GPT-4 for generating thought annotations. It can very well happen that miniF2F-related proof-steps are leaked in the thought generation process. Also, miniF2F has been used for a while for evaluating approaches for formal theorem proving, there is a chance that the base pre-trained model itself is trained on miniF2F repository. The worst part is that proofs are already checked in miniF2F repository, I would like to understand how much overlap exists between proofs generated by LeanStar and the proofs already checked in the miniF2F repository. Even the LeanDojo dataset, has existed in the public domain for quite some time, the pre-trained models like InternLM might as well be trained on this mix. I would like to see a broad study about how data leakage "impacts"/"doesn't impact" the overall results. I can also recommend some new benchmarks like Putnam Bench (https://arxiv.org/abs/2407.11214), which don't have proofs checked in online, for further evaluation.

It is a well-written paper, I will be more than happy to increase my score if the authors address some of my concerns.

**Questions:**

My main concern is the impact of data leakage (as highlighted in the weakness section) which needs to be studied further. The paper is well-written, I don't have any other questions.

---

> ### Author Response · Authors · 2024-11-24
>
> Thank you for your insightful review. We are glad that you appreciate our Lean-STaR method and found our paper well-written.
> ### Question 1:Data Leakage
> Our experimental setting (fine-tuning on Mathlib, evaluating on miniF2F) follows a widely used experimental setup in benchmark evaluations in neural theorem proving (e.g., InternLM was evaluated on miniF2F). Therefore, we believe that InternLM2-Math-base-7b has not been exposed to miniF2F. Also, GPT-4 is only used for synergizing the informal thoughts of the Mathlib dataset. It should not cause data leakage since problems in Mathlib and miniF2F are quite different.

---

> > ### Comment · Reviewer_QMzb · 2024-11-25
> >
> > While it may be reasonable to believe that the miniF2F repository was not directly leaked in InternLM's pre-training through a source like the GitHub, however, there are many proofs checked in the miniF2F repository which are also mentioned elsewhere on the internet. It is hard to ensure it was not accidentally leaked into the training data from somewhere else, especially because InternLM was trained much after the creation of miniF2F. It would be nice if the authors could at least do some form of analysis as to how many proofs are similar to the proofs checked-in in the miniF2F repository.
> >
> > You can refer to Section A.1.4 and Table 4 of https://arxiv.org/pdf/2310.04353, an analysis similar to that for LeanStar will help us better understand how many new proofs LeanStar could do that were not mentioned directly in the miniF2F repository.

---

> ### Author Response · Authors · 2024-11-26
>
> Thank you for your thoughtful comments！Following your suggestion, we do an analysis similar to the analysis in https://arxiv.org/pdf/2310.04353. The result is shown in the following table. From the following table, we can see that almost all proofs generated by LeanSTaR are different from the proofs mentioned in the miniF2F. If we set aside straightforward simple cases, there is only one proof generated by LeanSTaR is the same as the proofs mentioned in the miniF2F.
>
>
> |                             | Proofs found in miniF2F-test                  | Proofs found in miniF2F-test                               | Proofs found in miniF2F-test                  | Proofs found in miniF2F-test       |   Proofs found in miniF2F-test              |  Proofs found in miniF2F-test                                   | Proofs NOT in miniF2F | Total |
> |:-----------------------------:|:-----------------------------:|:-----------------------------:|:-----------------------------:|:-----------------------------:|:-----------------------------:|:-----------------------------:|:-----------------------------:|:-----------------------------:|
> |                             | Single-Tactic Simple Proofs            | Single-Tactic Simple Proofs           | Single-Tactic Simple Proofs                               | Two-Tactic Proofs |    Longer OR Complex Proofs            |  Total                     |       |               |               |
> | Tactics Used                | linarith                                     | norm_num                       | nlinarith         | two tactics      | > 2 tactics OR 1 tactic multi-args  |       |sorry                 |       |
> | Proof Count                 | 11                                           | 12                             | 2                 | 18               | 39                                  | 82                    | 162   | 244   |
> | Exact Match Successful Proof Count     | 2                                            | 4                              | 0                 | 0                | 1                                   | 7                     | 0     | 7     |
> | 1st Tactic Match Successful Proof Count| 2                                            | 4                              | 0                 | 0                | 1                                   | 7                     | 0     | 7     |
> | Distinct Successful Proof Count        | 5                                            | 7                              | 2                 | 14               | 19                                  | 47                    | 43    | 90 / 112 (80.36%)   |
> | Distinct Successful Proof Count ex Single-Tactic     |       -                                       |   -                             |         -          | 6                | 12                                  | 18                    | 20    | 38    |
> | All Successful Proof Count             | 11                                           | 12                             | 2                 | 17               | 22                                  | 64                    | 48    | 112   |

---

> > ### Comment · Reviewer_QMzb · 2024-11-26
> >
> > Thank you for the analysis. Based on your analysis it seems that most of the proofs found are different and the proofs generated are likely not the consequence of accidental leaks. Hence, I have increased my score. It will be nice if you can add this analysis in your final version.

---

> > > ### Author Response · Authors · 2024-11-27
> > >
> > > Thank you for your insightful feedback and strong support! We’re pleased our analysis was helpful and will include it into the final version of the paper.

---

### Official Review · Reviewer_kzhz · 2024-11-02

**Soundness:** 3
**Presentation:** 2
**Contribution:** 3
**Rating:** 6
**Confidence:** 4

**Summary:**

This paper introduces a framework called Lean-STaR for training LLMs that incorporates both informal reasoning in the form of natural language (called thoughts) with formal reasoning in the form of Lean code to improve a LLMs theorem-proving capabilities. More concretely, the paper proposes to construct a tactic predictor as p(tactic | proof context) = Expectation_{thought ~ p(thought | proof context)} p(tactic | thought, proof context). This contrasts with direct tactic prediction which directly construct p(tactic | proof context) as is done in previous works. To train the tactic predictor, the authors propose the following pipeline. First, they introduce retrospective rationale generation to create a "thought-augmented" dataset (called CoT dataset) which consists of tuples of (tactic, proof context, thought) where each (tactic, proof context) are taken from an existing Lean corpus (Mathlib) and each thought is generated by GPT-4 with a specially-crafted prompt. Second, they take InternLM2-Math-base-7b, a LLM that is pre-trained on Mathlib, and perform supervised fine-tuning on the CoT dataset to create a model called Lean-CoT. Third, the authors use expert iteration to construct a Lean-STaR model by fine-tuning the Lean-CoT model on a dataset of proof trajectories that it generates. The authors show that Lean-STaR outperforms direct tactic prediction on minif2f and LeanDojo benchmarks.

**Strengths:**

- The idea for constructing and training a tactic predictor p(tactic | proof context) = Expectation_{thought ~ p(thought | proof context)} p(tactic | thought, proof context) by introducing a latent variable in the form of thoughts is novel.
- There are numerous experiments.

**Weaknesses:**

The presentation of the paper could be improved and I have one major concern about soundness of the results (see questions for the authors).

- In general, I found the abbreviations and terminology of the paper to be confusing. For example, the text refers to the dataset as a thought-augmented dataset but it is abbreviated as CoT dataset. A model for direct tactic prediction is abbreviated as a SFT model. Lean-STaR is both the name of the framework and a model that is produced by the framework.
- Automatic theorem proving and interactive theorem proving are used interchangeably throughout the text even though they are not the same concept.
- The paper seems to argue that GPT-4 does not perform well in generating "correct rationale through few-shot prompting" (lines 196) and then proceeds to use GPT-4 to generate rationale for the CoT dataset.
- Figure 3 is not very clear for me. For instance, I would expect the Base Dataset to be fed into GPT-4 as opposed to directly into the CoT dataset. The Expert Iteration loop is also not clear. I would expect Expert Iteration from Lean-CoT to feed into Lean-STaR, and a separate arrow kind/color to indicate the usage of models in inference mode to generate datasets (also for GPT-4).
- Table 7 -> Table 1.

**Questions:**

- I believe that the learned distribution is p(tactic, thought | proof context) as opposed to  p(tactic | proof context) = Expectation_{thought ~ p(thought | proof context)} p(tactic | thought, proof context) as presented in the paper. In particular, I cannot find at any point in the paper how the expectation is handled (e.g., with a Monte Carlo estimate). Moreover, the dataset associates a single thought with a single (tactic, proof context) tuple so that we do not have multiple thoughts for a given proof context. Finally, my understanding of Figure 2 is that it is consistent with p(tactic, thought | proof context) since a single thought is generated and then used to generate a tactic as opposed to multiple thoughts that are marginalized. If so, it's not really a fair comparison with direct tactic prediction since you are basically comparing p(tactic | thought, proof context) vs. p(tactic | proof context) which means your approach gets strictly more information. In other words, the thought is not really treated as a latent variable and what is presented in the paper is not what is actually done. Looking forward to clarifying this.
- Is there an intuitive reason why sampling works better than best first-search for your method?

---

> ### Author Response · Authors · 2024-11-24
>
> Thank you for your insightful review of our paper. We're glad to hear that you appreciate our Lean-STaR method in formal theorem proving. We address your questions below.
> ### Concern 1: The abbreviations and terminology of the paper to be confusing. For example, the text refers to the dataset as a thought-augmented dataset but it is abbreviated as CoT dataset. A model for direct tactic prediction is abbreviated as a SFT model. Lean-STaR is both the name of the framework and a model that is produced by the framework.
> We understand the reviewer’s concern regarding the abbreviations and terminology. To clarify:
>
> 1) Following the convention in LLM post-training, the SFT model specifically refers to a model that is fine-tuned on the instruction-tuning dataset (i.e., Leandojo dataset in our paper). However, not all models designed for direct tactic prediction are SFT models. For example, a pretrained model that skips the SFT phase could still perform direct tactic prediction but would not qualify as an SFT model.
> 2) The term thought-augmented dataset refers to data that includes chain-of-thought (CoT) annotations. We abbreviate this as the CoT dataset.
> 3) Lastly, the Lean-STaR model is the product of the Lean-STaR framework. Although the same name is used for both, this follows a common practice in LLM research where frameworks and their outputs often share a name (e.g., RLHF algorithm and RLHF models, PPO algorithm and PPO models).
>
> We hope this clarification addresses the concern and aligns our terminology with established conventions in the field.
>
> ### Concern 2: Automatic theorem proving and interactive theorem proving are used interchangeably throughout the text even though they are not the same concept.
> We use “interactive theorem proving” (Section 3.1)  to mean a proving environment like Lean while we use “automatic theorem proving” to mean the task we solve. We hope this is clear.
> ### Concern 3: Figure 3 is not very clear for me. For instance, I would expect the Base Dataset to be fed into GPT-4 as opposed to directly into the CoT dataset. The Expert Iteration loop is also not clear. I would expect Expert Iteration from Lean-CoT to feed into Lean-STaR, and a separate arrow kind/color to indicate the usage of models in inference mode to generate datasets (also for GPT-4).
> Sorry for the confusion in Figure 3. Arrows there means we use some  model/dataset to produce a new model/dataset. For example, we use Base dataset and GPT-4 to produce CoT Dataset, and we use Lean-CoT model and Base dataset to produce Lean-STaR model. We will update the caption to make it more clear.
> ### Question 1: The learned distribution is p(tactic, thought | proof context) as opposed to p(tactic | proof context) = Expectation_{thought ~ p(thought | proof context)} p(tactic | thought, proof context) as presented in the paper. My understanding of Figure 2 is that it is consistent with p(tactic, thought | proof context) since a single thought is generated and then used to generate a tactic as opposed to multiple thoughts that are marginalized. If so, it's not really a fair comparison with direct tactic prediction since you are basically comparing p(tactic | thought, proof context) vs. p(tactic | proof context) which means your approach gets strictly more information.
> We would like to clarify that the thought (CoT) is treated as a strictly latent variable in our framework. For a given proof context, the model first generates a thought from the distribution p(thought | proof context), and then generates a tactic conditioned on both the thought and the proof context, using p(tactic | thought, proof context). The overall distribution of the tactic given the proof context is thus computed as p(tactic | proof context) = Expectation_{thought ~ p(thought | proof context)} p(tactic | thought, proof context), meaning it marginalizes over all possible thoughts. During evaluation, thoughts are dynamically generated for each proof context, which ensures that the process reflects the marginalization.
> Regarding fairness, both our method and direct tactic prediction rely solely on the proof context as input. The difference is that our approach explicitly models an intermediate reasoning step by generating thoughts before predicting tactics, while direct tactic prediction skips this step and directly maps the proof context to the tactic. This decomposition does not provide additional input information but introduces informal reasoning through the latent variable. Thus, the comparison is fair as both methods are ultimately conditioned on the same proof context.
> ### Question 2: Is there an intuitive reason why sampling works better than best first-search for your method?
> In Section 3.3, we claim that the sampling method performs better when using thought based models, and hypothesize that it may be because the average log probabilities used in the Best First Search method could be unreliable with the natural language-based CoT thoughts.

---

> > ### Comment · Reviewer_kzhz · 2024-11-25
> >
> > Thank you for your response.
> >
> > I am still not clear how your response addresses Question 1:
> >
> > > For a given proof context, the model first generates a thought from the distribution p(thought | proof context), and then generates a tactic conditioned on both the thought and the proof context, using p(tactic | thought, proof context). The overall distribution of the tactic given the proof context is thus computed as p(tactic | proof context) = Expectation_{thought ~ p(thought | proof context)} p(tactic | thought, proof context), meaning it marginalizes over all possible thoughts.
> >
> > In particular, the first sentence of the response is what I understood your framework to perform and is consistent with your figure. However, I do not understand how the second sentence follows and how marginalization is performed, which is the basis of my question. In particular, I would have expected discussion of how the marginal is approximated (assuming there is no closed form solution), e.g., by sampling multiple thoughts for a given proof context, to actually "marginalize over all thoughts". Can you help me understand more how this is done in your framework.

---

> > > ### Author Response · Authors · 2024-11-25
> > >
> > > Thank you for your thoughtful question. Let us clarify this:
> > >
> > > 1. The process of `proof context → thought → tactic` follows a probabilistic graphical model representation:
> > >    P(tactic, thought | proof context) = P(thought | proof context) * P(tactic | thought, proof context).
> > >    To sample from this multi-variate distribution, we sample the variables sequentially in the logical order (proof context → thought → tactic). This approach aligns with standard sampling methods for graphical models [see Slide 26 in [1]]. By doing this, the resulting tactic samples implicitly follow the marginal distribution:
> > >    P(tactic | proof context) = Expectation_{thought ~ P(thought | proof context)} [P(tactic | thought, proof context)].
> > >
> > > 2. There may be a misunderstanding in the need for marginalization. In our framework, we do not explicitly compute the marginalized value of P(tactic | proof context). Instead, we rely on unbiased sampling, which does not require knowledge of the exact marginalized probability. This eliminates the need to explicitly "marginalize over all thoughts." The sequential sampling approach ensures that our results are consistent with the desired marginal distribution without requiring the direct computation of the expectation.
> > >
> > > We hope this helps clarify the mechanism in our framework. Please let us know if further details would be helpful.
> > >
> > > [1] https://ics.uci.edu/~dechter/talks/tutorial-aaai-2010.pdf

---

> > > > ### Comment · Reviewer_kzhz · 2024-11-26
> > > >
> > > > Thank you for your response.
> > > >
> > > > I agree that you are sampling from
> > > >
> > > > A) P(tactic, thought | proof context) = P(thought | proof context) * P(tactic | thought, proof context).
> > > >
> > > > I do not agree that this is sampling from
> > > >
> > > > B) P(tactic | proof context).
> > > >
> > > > Can you please justify how to convert sampling P(tactic, thought | proof context) to sampling P(tactic | proof context) unless you marginalize P(tactic, thought | proof context) with respect to thoughts?
> > > >
> > > > Put another way, my understanding is that you are doing A in the paper but your paper is written/your understanding is that you are doing B. Indeed, Slide 26 that you reference is justification that you are doing A, not B, hence my major concern with this paper.

---

> ### Author Response · Authors · 2024-11-26
>
> We're happy to clarify any confusion further.
>
> > Can you please justify how to convert sampling P(tactic, thought | proof context) to sampling P(tactic | proof context) unless you marginalize P(tactic, thought | proof context) with respect to thoughts?
>
> When sampling $ (x, y) \sim P(X, Y \mid Z) $, the likelihood of obtaining a specific $ X = x $ is given by the law of total probability:
>
> $P(X = x \mid Z) = \sum_y P(X = x, Y=y \mid Z) \quad \text{(for discrete $Y$)},$
>
> or equivalently,
>
> $P(X = x \mid Z) = \int P(X = x, Y=y \mid Z) dy \quad \text{(for continuous $Y$)}.$
>
> This corresponds directly to the definition of the marginal distribution $ P(X \mid Z) $.
>
> Consequently, to draw a sample from $ P(\text{tactic} \mid \text{proof context}) $, one can simply sample from $ P(\text{tactic}, \text{thought} \mid \text{proof context}) $, and then discard the $\text{thought}$. This approach is exactly what we implemented in our method.

---

> > ### Comment · Reviewer_kzhz · 2024-11-26
> >
> > Thank you for your response.
> >
> > > Consequently, to draw a sample from P(thought | proof context), one can simply sample from P(tactic, thought | proof context), and then discard the thought. This approach is exactly what we implemented in our method.
> >
> > Note that by dropping thought, you are (implicitly) making an implementation choice on how to perform the marginalization and that there are other choices that could be made that could impact your algorithm's performance ... Nevertheless, I am satisfied with this formulation and appreciate your patience in clarifying this in our discussion. I hope that some aspects of this could be clarified/identified as a design decision of the paper, and also noted, possibly as an area of further investigation.

---

> > > ### Author Response · Authors · 2024-11-26
> > >
> > > Thank you for the insightful discussion and for raising these points. We appreciate your suggestions and will consider exploring this extension as one of our future research directions!

---

### Official Review · Reviewer_TwP5 · 2024-11-02

**Soundness:** 4
**Presentation:** 3
**Contribution:** 3
**Rating:** 8
**Confidence:** 4

**Summary:**

This paper introduces a novel approach to neural theorem proving (NTP). One previous drawback of step-by-step approaches to NTP is the lack of Chain-of-Thought (CoT) or SketchPad techniques before generating each tactic. These techniques have proven to be very useful in the domain of mathematical word problems (MWP). To address this, the paper proposes using GPT-4 to generate an initial CoT before each tactic on the Mathlib dataset, and subsequently bootstrapping the model with expert iteration. The experimental results show that adding CoT before generating the tactic leads to performance gains.

**Strengths:**

- The problem the paper aims to address is long-standing, and the paper provides strong empirical results demonstrating that the Chain-of-Thought (CoT) reasoning before each tactic application is useful.
- The newly proposed sampling method is novel and, based on its performance, appears to be a promising approach compared to Best-First Search.
- The proposed CoT-augmented dataset should be useful for future research in neural theorem proving.
- The paper is well-written and easy to follow.

**Weaknesses:**

- One issue with the generated COT is the absence of a verification mechanism or alternative methods to ensure its quality. The paper does not provide a detailed evaluation of these generated COTs; it only presents the final pass rate information. I believe that having a robust evaluation is crucial for improving the performance of this method. Moreover, in prompting GPT-4 to generate COTs, the authors have only explored one approach in constructing the prompt. I hope for a more in-depth analysis on how to create these prompts and a detailed examination of the structure of COTs that would better facilitate formal theorem proving.
- Another concern is that while the method demonstrates effectiveness, the ablation study reveals only a modest improvement of 2.5% (Lean-STaR versus Expert iteration (SFT) as shown in Table 3 in InternLM-Plus and 3.3% in InternLM-Base. Given that COT is expected to significantly enhance performance in the MWP (Math Word Problem) domain, I had anticipated a more substantial improvement. Could you provide any insights into why the performance of COT in NTP is not as strong as it is in MWP.

**Questions:**

- Are there any alternative methods employed in Lean-StaR to ensure the quality of the COT generated by GPT-4? Why was the state and ground truth tactic the sole approach used to generate these COTs? Would incorporating additional proving context improve the quality of the generated COTs?

- In Table 1, does the 'N' in 'Sampling' and 'Search' have different meanings? Is 'N' in 'Search' referring to the total number of expansions (nodes to explore), while 'N' in 'Sampling' refers to the number of failure retry attempts when the LLM generates a faulty tactic? I believe clarifying this in the caption would enhance understanding.

- In Lines 372 and 398, should 'Table 7' be corrected to 'Table 1'?

- It appears that the SFT finetune model does not show significant performance differences compared to InternLM2-7B. Is this due to the fact that these models have already been pre-trained on the Lean dataset?

- On Line 449, where does the 39.8% figure originate? Should it be 34.0% instead?

---

> ### Author Response · Authors · 2024-11-24
>
> Thank you for your constructive and positive feedback. We are glad that you appreciate our CoT-based method and dataset.
> ### Concern 1: COT is the absence of a verification mechanism or alternative methods to ensure its quality.
> Please see the  “**verification mechanism or filtering processGPT-4 used in CoT generation**” part under the General Responses. In short, we do have a filtering process for quality control of the GPT-4 generated COT in the STaR process.
>
> ### Concern 2: Why the performance of COT in NTP is not as strong as it is in MWP.
> In MWP without CoT, language models will directly give the answer. Therefore, part of the benefit of the CoT in our approach comes from that it lets the model think step by step. However, NTP is essentially step by step, so COT in NTP benefits only from combining informal thinking, which is not as strong as it is in MWP.
> A better analogy of MWP w/o CoT to NTP would be that the model sees the MWP problem, generates the equation for solving the problem, and the equation is symbolically solved by an external calculator
> ### Question 1: Are there any alternative methods employed in Lean-StaR to ensure the quality of the COT generated by GPT-4? Why was the state and ground truth tactic the sole approach used to generate these COTs? Would incorporating additional proving context improve the quality of the generated COTs?
> Please see the  “**verification mechanism or filtering processGPT-4 used in CoT generation**” part under the General Responses. In short, we do have filtering process for quality control of the GPT-4 generated COT in the STaR process.   We also agree with the reviewer that only using states and ground truth tactics is just the most direct and simple approach, and that Incorporating additional proving context is worth exploring in future research.
> ### Question 2: In Table 1, does the 'N' in 'Sampling' and 'Search' have different meanings? Is 'N' in 'Search' referring to the total number of expansions (nodes to explore), while 'N' in 'Sampling' refers to the number of failure retry attempts when the LLM generates a faulty tactic?
> 'N' in 'Search' refers to the total number of expansions as mentioned in Line 285 in our paper. However, 'N' in 'Sampling' refers to the number of all attempts (whether success or not). I.e., 'N' in 'Sampling' refers to the total number of tactics generated in one sampling process as mentioned in Line 293 in our paper.
> ### Question 3: In Lines 372 and 398, should 'Table 7' be corrected to 'Table 1'?
> Yes, that is a typo. Thank you for the feedback.
> ### Question 4: It appears that the SFT finetune model does not show significant performance differences compared to InternLM2-7B. Is this due to the fact that these models have already been pre-trained on the Lean dataset?
> Yes, InternLM model uses this leandojo training dataset in pre-training.
> ### Question 5: On Line 449, where does the 39.8% figure originate? Should it be 34.0% instead?
> Yes, that is a typo. Thank you for the feedback.

---

> ### Comment · Reviewer_TwP5 · 2024-11-24
>
> Thanks for your detailed explanations.
>
> Regarding my major concern, I think using the LLM to generate new tactics with the COT and testing their efficacy seems sufficient. Personally, I cannot think of another low-cost, effective method for this purpose. However, I am curious about the robustness of this filtering approach. Do you have any ablation study results that compare models trained with and without this filtering method? I believe including such an experiment in the paper would be valuable.
>
> For the other concerns and questions, your explanations have been adequate. I am now leaning towards acceptance.

---

> ### Author Response · Authors · 2024-11-25
>
> Thank you for your thoughtful comments and raising the score!
>
> To address your query, the COT model can be viewed as the version trained without the filtering method, as this "filtering" process is an integral part of STaR and not easily isolated. If you are interested in a comparison, we suggest referring to the results of LeanCOT versus LeanSTaR, which might provide the insight you’re looking for.
>
> We truly appreciate your feedback and are happy to clarify further if needed.

---

### Author Response · Authors · 2024-11-24

We are grateful for all the comments, questions and suggestions from the reviewers. Our responses are organized into two parts:
 1) the generic responses to the shared questions or suggestions from multiple reviewers, and
 2) the detailed responses to each point by individual reviewers.

Generic Responses

We are pleased to see the reviewers agree on the strengths of the paper, including:
 1) Focusing on an **important problem** (TwP5, MECo) in AI for math;
 2) Proposing a **novel approach** (MECo, kzhz, QMzb, TwP5) that combines the strengths of informal CoT reasoning, formal math proving with tactic generation and sampling-enhanced search;
 3) Providing **strong experimental results** (TwP5, QMzb, kzhz);
 4) Creating **a large new dataset with CoT-augmented math proving**, which should be useful for future research (TwP5, MECo).
 5) The paper is **well written** (TwP5, QMZb) but some terminology and wording need improvement (kzhz).

As for some shared questions or concerns, our Generic Responses are listed below:

### Is the proposed method lack of a verification or filtering for quality control of GPT-4 generated thoughts? (questioned by TwP5, MECo)

Our Clarification: The GPT-4 generated informal CoT thoughts are in natural language that is hard (or impossible) to verify directly. Instead, we filter out the lower-quality CoT in the STaR process by using LLM to generate tactics with CoT and checking whether the tactics can successfully prove the theorem. This process indeed can filter out the lower-quality thoughts and keep the ones that lead to correct tactics for further finetuning. In short, we do have an effective filtering process for quality control of the GPR-4 generated CoT. We will make this point more clear in our revised version of the paper.

---

### Meta-Review · Area_Chair_7z5x · 2024-12-23

**Metareview:**

This paper proposes a novel framework, Lean-STaR, which interleaves informal reasoning in natural language (or "thoughts") with formal reasoning in Lean (tactic prediction). The Lean-STaR framework combines several interesting techniques including LLM fine-tuning, self-taught reasoner, and expert iteration; although each idea is not new, the combination in theorem proving regarding leveraging informal reasoning is novel. The effectiveness of Lean-STaR has been shown convincingly through extensive experimental evaluations. Overall, Lean-STaR is an instance of a much general and promising framework with great potentials in theorem proving.

**Additional Comments On Reviewer Discussion:**

There were active discussions between reviewers and authors regarding both presentation clarifications, related work discussion, and some major concerns. Reviewer QMzb raises the concern of data leakage and suggests a simple study of assessing the risk. New results given by the authors indicate the concern of data leakage is less likely true. Reviewer MECo raise the slight concern of financial burden and potential of having multiple expert iterations. The authors clarify the financial burden is modest and two iterations is not yet the full potential of expert iterations, making the new frameworking more promising. Reviewer kzhz raises the concern that what is formally claimed in the paper (computing marginal) is not what is implemented (taking a single sample) in the paper, which can be a serious conceputal mismatch. The authors response clarifies a bit but is not fully satisfying. It appears that the implementation uses a single sample as the expectation, which in principle requires a more systematic treatment. The authors are encouraged to elaborate this further in the revision.

---

### Decision · Program_Chairs · 2025-01-22

Accept (Spotlight)